# Transmembrane Helices 7 and 8 Confer Aggregation Sensitivity to the Cystic Fibrosis Transmembrane Conductance Regulator

**DOI:** 10.3390/ijms242115741

**Published:** 2023-10-30

**Authors:** Bertrand Kleizen, Eduardo de Mattos, Olga Papaioannou, Michele Monti, Gian Gaetano Tartaglia, Peter van der Sluijs, Ineke Braakman

**Affiliations:** 1Cellular Protein Chemistry, Bijvoet Centre for Biomolecular Research, Utrecht University, 3584 CH Utrecht, The Netherlands; b.kleizen@uu.nl (B.K.); e.demattos@uu.nl (E.d.M.); o.papaioannou@uu.nl (O.P.); p.vandersluijs1@uu.nl (P.v.d.S.); 2Center for Life Nano- & Neuro-Science, Fondazione Istituto Italiano di Tecnologia (IIT), 00161 Rome, Italy; michele.monti@iit.it (M.M.); gian.tartaglia@iit.it (G.G.T.); 3Centre for Human Technologies (CHT), Istituto Italiano di Tecnologia (IIT), 16152 Genoa, Italy

**Keywords:** CFTR, aggregation, TMD2, TM7, TM8, instability, thermal sensitivity

## Abstract

The Cystic Fibrosis Transmembrane Conductance Regulator (CFTR) is a large multi-spanning membrane protein that is susceptible to misfolding and aggregation. We have identified here the region responsible for this instability. Temperature-induced aggregation of C-terminally truncated versions of CFTR demonstrated that all truncations up to the second transmembrane domain (TMD2), including the R region, largely resisted aggregation. Limited proteolysis identified a folded structure that was prone to aggregation and consisted of TMD2 and at least part of the Regulatory Region R. Only when both TM7 (TransMembrane helix 7) and TM8 were present, TMD2 fragments became as aggregation-sensitive as wild-type CFTR, in line with increased thermo-instability of late CFTR nascent chains and in silico prediction of aggregation propensity. In accord, isolated TMD2 was degraded faster in cells than isolated TMD1. We conclude that TMD2 extended at its N-terminus with part of the R region forms a protease-resistant structure that induces heat instability in CFTR and may be responsible for its limited intracellular stability.

## 1. Introduction

The cell surface is rich in membrane proteins such as receptors, cell adhesion molecules, transporters, and channels, which transmit signals from the outside world to the cell interior. Membrane proteins are critical for homeostasis and account for a major fraction of all drug targets [1,2]. A substantial fraction of membrane proteins traverse the lipid bilayer more than once. This complex architecture demands an extensive web of chaperones and cochaperones in topologically different locales to assist in folding into functional molecules. Many diseases originate from inherited defects in protein folding [3], including familial hypercholesterolemia [4], Nephrogenic Diabetes Insipidus [5], and neurodegenerative diseases [6]. The general theme for these conformational diseases is the aggregate formation of misfolded proteins induced by specific mutations.

Protein misfolding is also often the basis for Cystic Fibrosis, the most common genetic disorder amongst Caucasians. The mutation with the highest incidence, in ~90% of patients, is a deletion of phenylalanine at position 508 in the Cystic Fibrosis Transmembrane Conductance Regulator (CFTR) protein, a chloride channel functioning at the apical cell surface of epithelia [7]. Due to the F508 deletion, the mutant protein misfolds, is retained in the endoplasmic reticulum (ER), and degraded via proteasomes [8,9] and possibly autophagy [10,11].

Blocking proteasome-mediated degradation in cells overexpressing wild-type or mutant CFTR causes cytosolic accumulation of aggresomes, demonstrating that efficient protein degradation can either prevent aggregate formation or remove aggregates [12,13]. These earlier results suggest that both wild-type and mutant CFTR are inefficiently folding proteins. Other more recent studies in Calu-3 [14] and HEK293T cells [15], however, show that wild-type CFTR folds with an efficiency of approximately 80%. Whether the wild-type protein is an efficiently folding protein or not, CFTR clearly is extremely heat-intolerable and aggregates easily [16,17,18,19,20].

CFTR is a multi-spanning membrane protein (Figure 1A) that consists of two nucleotide-binding domains (NBD), a unique Regulatory (R) region lacking in other ABC transporters, and two membrane-spanning domains (TMD1 and TMD2). The ~200 amino acid-long R region is enriched in charged residues and links NBD1 and TMD2. NBD domains and R are in the cytosol and mediate channel opening and closing in a cAMP-protein kinase A-dependent manner. Both TMD structures, each with six hydrophobic transmembrane segments, anchor CFTR in the lipid bilayer and determine ion selectivity but also bury a substantial part of the protein in the membrane. Cryo-EM structures have become available for the phosphorylated and non-phosphorylated forms of CFTR, which show that the ion conductance pathway is formed by a wide cytoplasmic vestibule and a narrow transmembrane tunnel lined by positively charged amino acids, which tapers off in a gate close to the extracellular surface [21,22].

Much effort has been devoted to understanding how the F508 deletion causes instability of the protein [23,24,25]. Which region(s) in wild-type CFTR cause instability and aggregation is not known, however, in part because the lipid-embedded TMDs until recently evaded experimental determination of folding [26]. In the present study, we identified the aggregation-prone parts within CFTR. By analyzing the thermo-sensitivity of CFTR nascent chains of a series of C-terminal CFTR truncations and TMD fragments, we found that the TMD2 domain was prone to misfolding and aggregation. Part of the R-region formed a protease-resistant structure together with TMD2, which was more aggregation-sensitive than other protease-resistant CFTR structures.

## 2. Results

### 2.1. Aggregation Sensitivity of CFTR Nascent Chains

As a first indication for regions within the CFTR structure that tend to misfold, we examined the relation between nascent chain length and aggregation. In vitro translation of CFTR for various lengths of time in the presence of semi-permeabilized HT1080 cells as a source of ER membrane allowed the capture of CFTR nascent chains of increasing sizes in the process of chain elongation [27]. Low concentrations of digitonin selectively open the plasma membrane of the HT1080 cells but leave other intracellular membranes intact. As a consequence, the cytosol leaks out of the cells, which then is reconstituted with rabbit reticulocyte lysate and components required for the translation reaction [28]. The inclusion of ^35^S-methionine and -cysteine in the incubation leads to the production of radioactive CFTR [27].

Since CFTR is the only radiolabeled protein in the assay, we could omit immunoprecipitations for its detection. The advantage of this is that it ensures analysis of all forms of CFTR in an unbiased manner. Lysates containing translocated CFTR nascent chains were denatured in Laemmli sample buffer for 5 min at 37 °C, 65 °C, or 95 °C to induce aggregate formation. Forms of CFTR were then analyzed by SDS-PAGE and detected by fluorography (Figure 1B). Assessing thermo-sensitivity by SDS-resistant aggregation and SDS-PAGE is an established method to determine the stability of CFTR [16] and other multi-spanning membrane proteins such as VMAT [29], Na^+^/H^+^ exchanger [30] or *E. Coli* glycerol facilitator GlpF [31].

CFTR nascent chains synthesized within 10 or 15 min tolerated all temperatures and hardly showed accumulation into the high-molecular-weight species that are indicative of aggregation (Figure 1B). On longer exposures of the fluorograph (not shown), the high-molecular-weight species were not seen either. In contrast, extending synthesis time to 20–25 min produced CFTR nascent chains that were not full-length yet showed a marked temperature-dependent increase in aggregation (Figure 1B, bracket 1), which became more pronounced when CFTR reached its full-length form (Figure 1B, arrowhead). These results showed that C-terminal portions of CFTR are more sensitive to misfolding and aggregation than the N-terminal part. In accord, unfinished CFTR nascent chains ranging in length from ~25 kDa to ~75 kDa present at the later time points were completely resistant to heating (Figure 1B, bracket 2), whereas, in the same sample, the longer nascent chains and full-length monomeric CFTR disappeared and accumulated into aggregates (Figure 1B, arrow and bracket 1).

### 2.2. Aggregation Propensity Is Increased When TMD2 Is Present in the Nascent Chain

To identify elements in CFTR that cause its aggregation, we analyzed the thermo-sensitivity of CFTR variants truncated after each subsequent domain, as illustrated in Figure 2A. Upon in vitro translation of wild-type CFTR and CFTR variants truncated after TMD1 (E395X), NBD1 (D674X), R (E838X), and TMD2 (D1202X), their aggregation tendency was determined by exposing each protein to increasing temperatures like in Figure 1B. In agreement with the aggregation of the longer CFTR nascent chains (Figure 1B), we found that both D1202X and wild-type CFTR started to aggregate at 70 °C, coincident with the disappearance of monomeric forms (Figure 2B, arrowheads). After increasing the temperature to 95 °C, both D1202X and wild-type CFTR disappeared completely (Figure 2B, arrowheads) and accumulated into high-molecular-weight aggregates (Figure 2B, brackets). In contrast and again consistent with Figure 1B, the shorter truncations either were largely resistant to aggregation (E394X and D674X) or started to become vulnerable (E838X) (Figure 2B). Although some aggregates appeared after the 95 °C incubation, the monomeric forms remained present, in contrast to D1202X and wild-type CFTR. Even though the shorter truncations already include six transmembrane segments (in TMD1), the variants up to and including the R region (the N-terminal 837 residues) showed little, if any, aggregation propensity.

### 2.3. Aggregation Propensity of TMD1 and TMD2 Is Independent of Other Domains in CFTR

The different aggregation behavior of truncations containing TMD1 and TMD2 could be caused by the architecture of CFTR, i.e., the position of each TMD in CFTR and its interaction with other domains. We, therefore, examined the aggregation propensity of isolated TMD1 (E395X) and TMD2 (E838-D1201) without the context of the other domains. To verify that TMD2 contains a sufficiently strong ER-targeting signal for translocation, we capitalized on the property of endo H to remove oligomannose-type N-linked glycans from glycoproteins. The fourth extracellular loop (ECL4) of CFTR contains N-linked glycosylation sites at positions 894 and 900 in TMD2. Since the in vitro translation–translocation reaction does not reconstitute ER-to-Golgi transport, consistent with the absence of a complex glycosylated form of TMD2, the glycans in TMD2 will remain endo-H sensitive. Indeed, digestion of the translation reaction with endo H converted the larger of the two TMD2 products quantitatively into the faster migration band (Figure 2C); this demonstrated proper glycosylation and insertion into the ER membrane of isolated TMD2.

After in vitro translation in the presence of semi-permeabilized cell membranes, isolated TMD1 (E395X, arrowhead) and TMD2 (E838-D1201, asterisk, and filled circle) were solubilized and subjected to increasing temperatures. Consistent with our previous findings, incubation of samples at 95 °C reduced monomeric full-length TMD2 in the gel and increased the number of aggregates (Figure 2C, bracket), whereas TMD1 remained monomeric (Figure 2C, arrowhead). Glycosylated (Figure 2C, asterisk) and non-glycosylated TMD2 (Figure 2C, filled circle) aggregated to the same extent (Figure 2C, bracket) at 95 °C, indicating that glycosylation did not protect TMD2 against aggregation in these experimental conditions.

Although TMD2 was intrinsically more prone to aggregation than TMD1, isolated TMD2 tolerated the 70 °C incubation (Figure 2C) to some extent; this differed from the aggregation sensitivity of full-length CFTR (Figure 2B), suggesting that other domains in the full-length protein contribute to the aggregation sensitivity of TMD2. When the N-terminal and C-terminal halves of CFTR are co-expressed in HeLa cells, the two TMDs can interact [32]. The ER membranes prepared from plasma membrane permeabilized cells collapse around the nucleus but do not fragment and, in principle, allow TMD1 and TMD2 produced by distinct polysomes to associate. We translated both TMDs in the same reaction to analyze whether the aggregation of either TMD was changed. Neither TMD1 solubility nor TMD2 aggregation turned out to be affected appreciably (Figure 2C), showing that the TMDs did not interact in this system or that other CFTR regions than TMD1 contributed to the aggregation of TMD2.

### 2.4. Identification of Protease-Resistant Aggregation-Prone R-TMD2 Fragments

Limited proteolysis is a well-established tool to probe conformation and identify folded domain structures within a protein [33,34,35,36]. We and others extensively used it to study the folding of de novo synthesized CFTR in in vitro translations [37,38] and cells [15,26]. The in vitro translation–translocation assay allows the examination of all CFTR domains [27]. To identify regions of CFTR that affect TMD2 folding and stability, we compared trypsin digestion patterns of TMD2 alone (E838-D1201) with that of several C-terminal CFTR truncations and wild-type CFTR (Figure 2A and Figure 3A). D674X and E838-D1201 were translated for 30 min, and E838X, D1202X, and wild-type CFTR for 60 min in the presence of semi-permeabilized cells. All CFTR constructs were translated into full-length protein and analyzed (Figure 3A, left panel). Below each protein, nascent chains were detectable, like for wild-type CFTR. After synthesis, proteins were solubilized and subjected to 5 μg/mL or 25 μg/mL trypsin treatment to determine protease susceptibility. Of note, the proteolytic patterns of all three C-terminal CFTR truncations resembled that of full-length CFTR, suggesting that they were inserted into the membrane and acquired native-like states. The few differences allowed the identification of many of the protease-resistant fragments: the addition of a domain will add bands specific to that domain or will change bands derived from the C-terminus of the preceding domain (Figure 3A, middle, and right panels).

Trypsin digestion of D1202X and wild-type CFTR showed a specific protease-resistant fragment of ~42 kDa (Figure 3A, middle and right panels, filled circle) that was absent in CFTR truncated already after the R-region. This protease-resistant ~42-kDa structure was invariant even upon the addition of five times more trypsin (Figure 3A, right panel) and contained at least part of TMD2 but no residues downstream of 1201. The slowest migrating protease-resistant fragment of TMD2 (E838-D1201) was barely smaller than undigested TMD2, ~35 kDa (Figure 3A, open circle) and appreciably smaller than the 42 kDa fragment. We concluded that the ~42 kDa protease-resistant structure contained parts of TMD2 and the upstream, neighboring R-region.

When proteolytic fragments of wild-type CFTR generated with 25 µg/mL trypsin were subjected to thermal denaturation, the 42 kDa R-TMD2 band became fuzzier, and part of it accumulated in aggregates on top of the gel (Figure 3B, bracket). Almost all smaller protease-resistant fragments resisted aggregation at 95 °C (Figure 3B), showing that the R-TMD2 structure represents a weak point in the stability of CFTR.

To establish whether complete TMD2 was part of the 42 kDa fragment, we analyzed wild-type CFTR and CFTR variants truncated after R (E838X), after ICL3, intracellular loop 3 (L998X)), or at the end of the cytosolic helix extending from TM12 in TMD2 (K1165X), translated in vitro for 60 min as above. Samples were subjected to limited proteolysis with 5 µg/mL trypsin to probe conformation and identify folded domain structures (Figure 3C, left panel). Fragments were immunoprecipitated with antibody G449, which was raised against R, residues 645–835 [39] (Figure 3C, middle panel). All detected polypeptide species, monomers, aggregates, and protease-resistant fragments, therefore, included part of R (Figure 3C). The 42 kDa fragment (filled circles) only arose from CFTR variants that contained full TMD2, suggesting that the 42 kDa fragment included TMD2 beyond residue 1164. The alternative explanation that the protease-resistant structure required those downstream residues for a compact fold was unlikely as K1165X was digested into only slightly smaller proteolytic fragments of ~39 kDa (Figure 3C, open circle); this implied that the ~39 kDa fragment originating from K1165X was a C-terminally truncated version of the 42 kDa fragment derived from D1202X and wild-type CFTR. Truncating TMD2 even further to L998X decreased TMD2 protease resistance, yielding at least a partial R-containing fragment of ~20 kDa (Figure 3C, hashtag) in addition to the fragments derived from E838X.

Further evidence for the identity of the 42 kDa fragment arose from digestion with endo H to probe for the presence of glycans. As expected, the fragment contained the TMD2 glycans between TM7 and TM8 and shifted down in the gel (Figure 3C, asterisk). An additional immunoprecipitated band of ~30 kDa was sensitive to endo H as well (Figure 3C, open arrowhead), showing that this fragment also contained both R and TMD2 parts. This fragment had similar electrophoretic mobility but was different from a fragment originating from E838X, as the latter lacks TMD2. The digest of the L998X truncation may well contain both fragments, one of which then moves up to ~39 kDa and 42 kDa in the digests from the larger CFTR variants (middle panel). We concluded that the 42 kDa fragment contained the complete TMD2 and a portion of R.

Because the difference in apparent molecular weight between isolated TMD2 (~35 kDa) and the R-TMD2 structure (~42 kDa) was less than the predicted molecular weight of the R-region (~17 kDa), the 42 kDa fragment is unlikely to contain complete R. Despite the 22 predicted trypsin cleavage sites in R, half of which in the C-terminal half, it remained resistant to the protease, consistent with a stable fold between this C-terminal part of the R-region and TMD2, which required at least part of TMD2.

To determine whether the results can be extended beyond in vitro conditions, we analyzed the thermal aggregation of CFTR produced in the cellular context. HEK293T cells expressing wild-type CFTR or variants truncated after R (E838X) or TMD2 (Y1219X) were pulses labeled for 15 min with ^35^S-methionine/cysteine (Figure 3D). Detergent lysates were subjected to limited proteolysis with 1 µg/mL trypsin (or not) and full-length proteins (top panel) or proteolytic fragments, then were immunoprecipitated with the G449 antibody against R and resolved by SDS-PAGE. During this short period of labeling, negligible amounts of newly synthesized CFTR molecules have already traveled to the Golgi complex, and that is why only the oligomannose forms are visible for wild-type and Y1219X CFTR (top panel), both eventually transported to the Golgi complex [40,41]. Truncation after R creates a half-CFTR molecule that cannot assemble its domains and, therefore, does not fold properly [42] and gets retained in the ER. Consistent with the results of the in vitro experiments, raising the denaturation temperature to 95 °C decreased the amount of native wild-type and of Y1219X, the latter only lacking NBD2 (Figure 3D, upper panel). The truncation after R (E838X, without TMD2 and NBD2) showed little, if any, aggregation propensity at 95 °C (Figure 3D, upper panel). Since CFTR was more sensitive to trypsin in detergent lysates from cells than in in vitro translation reactions, we titrated the trypsin concentrations. Limited proteolysis with 1 µg/mL trypsin followed by immunoprecipitation with G449 also recapitulated the in vitro results (Figure 3A–C) and uncovered bands at ~42 kDa for wild-type CFTR and Y1219X, which were diminished when samples were denatured at 95 °C (Figure 3D, middle and bottom panels).

### 2.5. Aggregation Propensity Is Distributed along the Entire TMD2 Sequence

To determine more precisely which part of TMD2 features the thermosensitive elements, we made use of a kinetic assay to analyze the folding of CFTR in vivo. The assay (as in Figure 3D) relies on a radiolabel pulse-chase—proteinase K limited proteolysis approach followed by immunoprecipitation of proteolytic fragments with antibodies against the individual CFTR domains [26], similar to in Figure 3D. The assay can discriminate between co-translational and post-translational CFTR folding because precisely defined proteolytic fragments generated from the TMDs increase in size during the de novo folding trajectory (Figure 4A) [26].

Wild-type CFTR-expressing HEK293T cells were radiolabeled for 15 min with ^35^S-methionine/cysteine, and the label was chased for 1.5 h or not (0 h chase). Immunoprecipitation with an antibody against TMD1 followed by thermal denaturation showed that both CFTR forms were prone to aggregation (Figure 4B, top panel): the ER form (the predominant species after 15 min labeling) and the Golgi form that arose during the chase and which has left the ER, has acquired the Golgi-localized modifications of its glycans and will be present not only in the Golgi complex but also on the plasma membrane and in endosomes. Immunoprecipitation of CFTR fragments from proteinase-K digested lysates with TMD2-recognizing antibodies showed that the T2a and T2b fragments that already arise from cotranslationally folded CFTR (0 h chase) were not sensitive to heat denaturation (Figure 4B, bottom panel). In contrast, the amount of monomeric T2c fragment, which is generated from CFTR that has assembled its domains into a mature structure [26], was reduced at 95 °C (Figure 4B). The N-terminus of T2c is residue 910 in the fourth extracellular loop, while T2a and T2b start at aa 1060 and 965, just before cytoplasmic coupling helices 3 and 4, respectively. The three fragments have the same C-terminus within the TMD2-NBD2 boundary at aa 1191 (Figure 4A) [26]. We therefore concluded that the major thermo-sensitive determinant in TMD2 of CFTR does not include the 4 C-terminal TMs in TMD2 but is a region encompassing TM7 and TM8 (Figure 5A, blue and black). Transmembrane helix TM8 represents the difference between proteinase-K-generated T2c and T2b, and the 42 kDa fragment in the trypsin digest suggests that the upstream helix TM7 is likely involved as well.

We next assessed aggregation propensity in silico [44]. Using the algorithm in combination with AlphaFold2.0 [47] and the cryo-EM structure of CFTR (PDB 5UAK) [43] confirmed that the major aggregate-prone parts are TM7 and TM8 (Figure 5B), with minor contributions of TM9-11 in the folded structure. In contrast, for TMD1, which has a comparable hydrophobicity profile as TMD2 (Figure 5B), only minor aggregation-prone regions are predicted for the folded state in accordance with the experimental data (Figure 2 and Figure 3). Since Alphafold predictions of unstructured regions have low confidence [48,49], we also computed aggregation propensity based on sequence alone and arrived at the same conclusion (not shown). Thus, theoretical and experimental data show that TM7 and TM8 form the aggregation-sensitive elements in CFTR).

### 2.6. TMD2 Has a Higher Turnover Rate and Is Less Stable than TMD1 In Vivo

Having established that TMD2 and not TMD1 is the unstable domain in CFTR, we set out to assess whether this translated to the physiological context. To determine degradation kinetics in cells, we transiently expressed isolated TMD1 and TMD2 domains in HEK293T cells, radiolabeled these for 5 min, and chased them for different periods. The chase was conducted in the presence of 1 mM cycloheximide to stop protein synthesis after pulse labeling and synchronize the entry of full-length domains into the chase period. The TMD proteins were then immunoprecipitated from detergent lysates and analyzed by SDS-PAGE. To facilitate the detection of TMD2, we employed a TMD2 construct with a triple-HA tag in ECL4 as pioneered by the Lukacs lab [50]. A comparison of TMD1 and TMD2 turnover showed that TMD2 is degraded faster than TMD1 (Figure 6), in agreement with our earlier measurements [40]. These data show that the in vitro thermostability of TMD1 and TMD2 correlates well with their cellular stability, suggesting that aggregation-sensitive TMD2 is recognized by molecular chaperones as an unstable domain and triaged for ER-associated degradation.

## 3. Discussion

The multi-spanning membrane protein CFTR often misfolds in cells and is prone to aggregation. Many large membrane proteins are known to form SDS-insoluble aggregates upon heat treatment. We found that aggregation of CFTR was highly specific for structure and sequence: the seventh and eighth transmembrane helices in TMD2 (Figure 5A, blue and black) caused CFTR aggregation, whereas, in TMD1, the cytosolic domains and the N-terminal half of CFTR (including TMD1, NBD1, and R) did not. Part of the R-region formed a protease-resistant, folded, aggregation-sensitive structure with TMD2. We concluded that TMD2 is the source of CFTR’s instability.

Although our results were obtained in a denaturing environment and in part in vitro, they are relevant for the folding of a protein in its natural cellular environment where polypeptides will continuously be sampling folding intermediate conformations and interacting with other parts of CFTR and with chaperones. Three lines of reasoning support this relevance.

First, we already showed that in cells, TMD1, NBD1, and TMD2 fold already during the translation of CFTR into a similar conformation as when expressed in isolation [26,40]. We have learned to distinguish domain folding from domain assembly [26]. During assembly, TMD1, TMD2, and NBD2 acquire increased protease resistance. Studies with a synthetic construct in which TMD1 and TMD2 are directly joined revealed that this protein can be exported out of the ER to the Golgi complex [51].

Second, whereas SDS denatures a protein and indeed breaks its conformation, a protein forms alternative structures together with SDS and rarely ends up linear [52]. We established that even 3% SDS is insufficient to completely unfold CFTR and failed to find conditions where proteases would completely digest CFTR. As an alternative, we then chose a misfolded mutant (CFTR lacking R) to demonstrate that the discrete fragments we found depended on CFTR being folded [15].

Third, while considering extrinsic factors such as solvent, buffer, or temperature, our in silico analysis uncovers intrinsic contributions that are sequence-dependent; this enabled the identification of regions that are either prone or resistant to aggregation and supported the experimental data.

Fourth, the results and conclusions on heat sensitivities we describe in the manuscript are consistent with the cell biological stability (Figure 6), in accordance with the seminal work of Wittrup and associates [53,54], who established that cell biological stability correlates well with thermal stability.

Both the in silico and biochemical analyses uncovered the aggregation propensity of TMD2 but not TMD1. Even more specifically, the presence of only TM7 and TM8 rendered CFTR variants unstable, whether extended at the N-terminus –with R region– or at the C-terminus –in TMD2–. TM11 displayed in silico a larger aggregation propensity as well, but only in a completely unfolded state, which exists neither in the cell nor in our experiments. Neither TM11 nor TM7-TM8 have been reported as unstable or aggregation-prone regions in wild-type CFTR. Instead, two other structural elements of CFTR have been pinpointed in NBD2 and TMD1. A distinct C-terminal region of 110 amino acids in NBD2 confers aggregating propensity when fused to GFP [55,56,57]. Its significance for stability in the context of full-length CFTR has not been shown, however, and these 110 residues may well require the rest of NBD2 to acquire a native, more stable fold. Our in silico analysis revealed that NBD2 in both folded and unfolded states has little, if any, propensity to aggregate. NBD2, nevertheless, is not stable. It requires other domains in CFTR to acquire a folded, protease-resistant state [58] and solubilizing mutations to allow purification and in vitro studies [59]. The unstable feature in TMD1 is of a different nature: its most C-terminal transmembrane helix, TM6, contains three positively charged amino acids that are important for CFTR function as a chloride channel but hamper integration of TM6 into the ER membrane together with its relatively short length [60]. TM6 has a predicted ∆G of 4.76 kcal/mol for membrane insertion (https://dgpred.cbr.su.se/index.php?p=home, accessed on 9 May 2023), which makes it the predicted most unstable TM helix of CFTR [61]. Experimental evidence as a fusion protein with invertase shows that it can contribute to inefficient maturation [58].

The 42 kDa trypsin-resistant fragment from wild-type CFTR consists of at least part of the R-region folded into a compact structure with TMD2, even though isolated R is highly polar, contains sufficient cleavable residues, is intrinsically disordered, and is not aggregation-prone. In purified form, it forms a random coil conformation [62], and it remained largely unresolved in cryo-EM structures of CFTR from zebrafish [21], humans [43], and chickens [63]. The R region lodges between the two NBDs and keeps them apart in the inactive state [21]. Through phosphorylation, R pivots out of this position, allowing the NBDs to dimerize and form an open channel. The phosphorylation status of R regulates association with an increasing number of identified partners, which we showed here to include TMD2. Our results extend co-expression and NMR studies demonstrating that the R-region physically and functionally interacts with upstream and downstream CFTR domains: the N-terminus, NBD1, NBD2, ICL3 in TMD2, and the C-terminus [64,65,66,67,68,69,70]. The presence of the R-TMD2 fragment in trypsin digests from both cell lysates and in vitro translations suggest it to represent a bona fide folded structure within CFTR. This is reminiscent of the N-C-terminal interactions with ICL2 in TMD1 [43], as part of R represents the N-terminus of TMD2. The increased protease resistance of the 42 kDa R-TMD2 fragment in absence of NBD2 (in Y1219X) compared to wild-type may well imply a decrease in packing of TMD2 upon binding of NBD2 to ICL3.

Since the hydrophobicity predictions for TMD1 and TMD2 were similar, additional biophysical properties of TMD2 must explain its aggregation propensity. TMD1 and TMD2 were similar also in distribution and numbers of positively-charged residues, negatively-charged residues, and net charge. Protein size alone clearly did not trigger the aggregation of CFTR because both isolated TMD2 (366 residues) and the R-region/TMD2 fragment were aggregation sensitive, whereas the larger E838X variant did not aggregate. Aggregation propensity must have resulted from the amino acid sequence, TM length [60,71], and structure. The minimal part of CFTR that triggered aggregation included TM7 and TM8 with the connecting ECL4.

During co-translational folding and membrane integration of CFTR, the ribosome needs to engage again with the translocon for insertion of the TMD2 TM segments after translating the large cytosolic bulk of NBD1 and R. Correct translocation of the first TM segments of TMD2, TM7 and TM8, is crucial for proper CFTR folding and insertion [72]. The two N-linked glycans located on asparagine residues in positions 894 and 900 in ECL4 between TM7 and TM8 are added efficiently and likely contribute to the stable insertion of TM7 and TM8. Because non-glycosylated CFTR is not detectable in cells nor in vitro, TM7 and TM8 must have been properly translocated in both systems. TM7 can function as a signal peptide and is the most hydrophobic transmembrane helix of CFTR with a predicted ∆G for insertion −3.36 kcal/mol, ensuring strong anchoring in the bilayer during synthesis. In contrast, TM8 is not stably integrated into the ER membrane and needs TM7 for proper translocation and insertion [72,73].

Cryo-EM structures have demonstrated that TM8 is not a typical continuous transmembrane helix but is discontinuous and makes two breaks within the bilayer [21] (Figure 5A, black). These breaks free main chain atoms from their secondary structure allowing them to interact with solutes and water along the pore, facilitating gating [21]. For CFTR activity as a regulated channel, TM8 is important, but such dynamics endanger protein stability. Moreover, the charged aspartate D924 located at the breaks (Figure 5A, grey spheres) has a high propensity to be in a loop and reduces TM8 hydrophobicity. The unstable structure TM8 is stabilized, however, by contacts with the other unstable transmembrane helix TM6 in TMD1, which are critical for the architecture of the channel and the narrow selectivity region of the channel [74,75]. D924 forms a salt bridge in the membrane with R347 in TM6; R347P is a functional mutant found in Cystic Fibrosis patients. Conversely, the polar Asp924 and Arg347 can increase the overall stability of TM6-8 segments within the membrane protein by inducing strong SDS-resistant inter-helical hydrogen bonds that are favored in the apolar milieu of the bilayer, as competing water is largely absent [76,77]. Polar residues in aberrant TM segments, however, can induce strong undesirable hydrogen bonds between TM helices that lead to the formation of oligomers [78,79] and possibly larger aggregates ([80] and refs therein). Indeed, an earlier study showed that recombinant isolated TMD2 can homodimerize and oligomerize in vitro [81].

The first hurdle to proper CFTR folding and maturation is the proper insertion of TMD1 and TMD2 into the ER membrane. An increasing number of translocons is being identified next to the major, conserved Sec61 translocon [82], GET [83], EMC [84], TMCO [85], SND [86], GEL-BOS-PAT complex [87,88]. TMD2 has more difficulty acquiring a stable structure than other CFTR domains and, therefore, may need more assistance from cellular components to reduce its tendency to aggregate. Translocons may be considered molecular chaperones for the integration and stabilization of intramembrane segments but may support CFTR folding after translation as well. Several specialized folding factors in the ER membrane have been shown to associate with CFTR, but clarity on their mechanism of action is lacking. An example is the B-cell antigen receptor, for which the transmembrane domain of mIgD binds BAP29 and BAP31 [89]. BAP31 also interacts with TMD1 of CFTR and reduces cell surface expression of CFTR, whereas knockdown rescues CFTR-F508del [90,91]. By acting on TMD1, BAP31 may indirectly contribute to chaperoning TM segments of TMD2 as well. Another example is the transmembrane lectin calnexin, one of the most abundant ER chaperones and a key regulator of glycoprotein quality control [92]. Calnexin interacts with CFTR through the N-linked oligosaccharides in ECL4 [93,94], promotes interactions between TMD1 and TMD2 [95] and folding of CFTR [96]. Calnexin also binds to and chaperones folding of multipass membrane proteins that are not glycosylated [97,98] and indeed was shown to interact with many non-glycosylated transmembrane proteins via their transmembrane domains [99,100]. Perhaps intramembrane client recognition by calnexin contributes to CFTR folding by preventing TMD2-dependent aggregation.

Of note, the potentiator drug VX-770 binds to a cleft formed by TM4, TM5, and TM8 of human [101] CFTR. This site coincides with the hinge region in TM8, a structural feature of CFTR not found in other ABC transporters [101], including thermostabilized chicken CFTR [63]. Nevertheless, VX-770 also acts on ABCB4 [102], ABCB11 [103], and ABCG2 [104], suggesting that features of TM4 and TM5 in the binding site may be retained in other ABC transporters. Alternatively, since the potentiator GLPG1837 acts through the same mechanism as VX-770 [105] and molecular docking identified a second binding site for GLPG1837 involving TM12 [106], the other ABC transporters may be targeted via this second site.

Altogether, our results show that C-terminal parts of CFTR, in which TMD2 is the major determinant, are more prone to misfolding and aggregation. Interestingly and consistent with its stability in our experiments, CFTR truncated after the R-region (1-836) generated functional channels at the cell surface [107] that were degraded even less than wild-type protein [108,109]. It seems that driven by evolution, the conductance properties of CFTR became more fine-tuned with the addition of TMD2 and NBD2, at the price of TMD2 being a source of instability in CFTR.

## 4. Material and Methods

### 4.1. Cell Lines and Transfection

HT1080 and HEK293T cells were cultured as described [26,27]. HEK293T cells were seeded on poly-L-Lysine (Sigma, St. Louis, MO, USA) coated dishes and transfected 24 h before experiments, using 40 kDa polyethyleneimine (PEI; Polysciences, Warrington, PA, USA). DNA/PEI mixtures (ratio 1:3 (*w*/*w*) with 12.5 mg PEI for a 6 cm dish) were pre-incubated in 150 mM NaCl for 20 min at room temperature, and PEI-DNA complexes were then added to the cells. After 4 h, transfection mixtures were aspirated and replaced with growth medium, and cells were cultured for 16–20 h prior to experiments in humidified incubators at 37 °C under 5% CO_2_.

### 4.2. Expression Constructs and Antibodies

pBS-CFTR(E395X), pBS-CFTR(D674X), pBS-CFTR(E838X), pBS-CFTR(K1165X), pBS-CFTR, pBS-CFTR(E838-D1201), pBI-CMV2-CFTR, pBi-CMV2-CFTR(D674X), pBi-CMV2-CFTR(E838X), pBi-CMV2-CFTR(E838-D1201)-HA3 have been described [15,26,27,40]. pBS CFTR(L998X), pBS CFTR(D1202X), and pBi-CMV2-CFTR(Y1219X) were generated with PCR using pBS-CFTR as template and primers from Appendix A. All constructs were sequence verified. Rabbit antibodies MrPink against NBD1, G449 against R, E1-22 against TMD1, TMD2C against TMD2, and MrBrown against HA have been described [26,110], and their epitopes are detailed in Appendix A.

### 4.3. Preparation of Semi-Permeabilized HT1080 Cells

The preparation of semi-permeabilized cells as a source of ER membrane was described before [28]. In brief, one 10 cm dish of 80% confluent HT1080 cells was trypsinized and resuspended in 9 mL ice-cold 110 mM KOAc, 20 mM HEPES pH 7.2, 2 mM MgOAc_2_ (KHM) containing 10 µg/mL Soybean Trypsin Inhibitor (Sigma). All subsequent steps were at ice temperature. Cells were spun at 250× *g* for 3 min and resuspended in 6 mL ice-cold KHM. The plasma membrane was selectively opened with 40 µg/mL digitonin (Calbiochem, San Diego, CA, USA), and permeabilization was stopped by adding 8 mL ice-cold KHM. Cells were immediately spun down, resuspended in 10 mL ice-cold 50 mM HEPES pH 7.2, 90 mM KOAc, and kept for 10 min on ice. Cells then were centrifuged, and the pellet was resuspended in 1 mL ice-cold KHM. Permeabilization efficiency was assessed microscopically with Trypan Blue (Fluka, Munich, Germany) staining. Next, semi-permeable cells were centrifuged for 15 s at 10,000× *g* and resuspended in 0.1 mL KHM. Endogenous mRNA was removed by incubation with 1 mM CaCl_2_ and 10 µg/mL micrococcal nuclease (GE Healthcare, Chicago, IL, USA) for 12 min at room temperature. Micrococcal nuclease was inactivated with 4 mM EGTA, and permeabilized cells were retrieved by centrifugation, resuspended in KHM, and used in the in vitro translation–translocation assay.

### 4.4. In Vitro Translocation–Translocation

mRNA transcription was conducted as described [27], and mRNA was dissolved in 50 µL nuclease-free H_2_O containing 20 units of SUPERase. In RNAse inhibitor (Ambion, uden, The Netherlands) and stored at −80 °C. In vitro translation–translocation in the presence of semi-permeabilized HT1080 cells has been described [28,111]. In brief, 10 μL translation mix, which corresponds to one sample, contains 6.6 μL Flexi^®^ rabbit reticulocyte lysate (Promega, Leiden, The Netherlands), 1 mM amino acid mix without methionine and cysteine, 50 mM KCl, 1 mM DTT, 2.1 μL freshly prepared semi-permeabilized HT1080 cells, 0.4 μL mRNA (~1 mg/mL), and 4 μCi ^35^S methionine and cysteine (10 μCi/μL, Redivue PRO-MIX L-[^35^S] (GE Healthcare, Chicago, IL, USA). Translation reactions were conducted at 30 °C for the indicated periods and terminated with 0.25 mL ice-cold 110 mM KOAc, 20 mM HEPES pH 7.2, 2 mM MgOAc_2_ (KHM) containing 1 mM cycloheximide. Reactions were transferred to ice, washed once (2 min at 10,000× *g* at 4 °C) with 0.75 mL ice-cold KHM, and resuspended in 20 μL KHM, 1% Triton X-100 (KHM lysis buffer). Lysates were then subjected (or not) to limited proteolysis (see below) and received an equal volume 2x reducing Laemmli sample buffer (final concentrations 200 mM Tris-HCl pH 6.8, 3% SDS, 10% glycerol, 1 mM EDTA, 0.004% bromophenol blue) and thermally denatured for indicated times and temperatures. For some experiments, full-length CFTR produced in vitro and proteolyzed peptides were immunoprecipitated before denaturation. Samples containing non-proteolyzed CFTR or limited proteolysis fragments were resolved on 7.5% and 12.5% SDS-PAA gels, respectively. Gels were analyzed on A Typhoon Imager and exposed to Kodak MR BioMax film.

### 4.5. Radioactive Pulse Chase Assay

Cells were starved for 15 min in methionine/cysteine-free MEM (ICN biomedicals, Aachen, Germany) and pulse-labeled with ^35^S-methionine and cysteine (125 μCi/mL). Radiolabeling was stopped after 15 min by adding excess, unlabeled 5 mM methionine and 5 mM cysteine. Cells were then chased (where indicated in the presence of 1 mM cycloheximide). At different chase times, cells were washed twice with ice-cold Hanks’ balanced salt solution (HBSS) (Life Technologies, Bleiswijk, The Netherlands) and solubilized in ice-cold 20 mM MES, 100 mM NaCl, 30 mM Tris-HCl pH 7.4 (MNT), 1% Triton X-100 without protease inhibitors. Part of the lysates was subjected to limited proteolysis (cf below), and CFTR and proteolytic fragments were immunoprecipitated with indicated antibodies. Samples were analyzed using 10% SDS-PAGE.

### 4.6. Limited Proteolysis

Cleared lysates were subjected to limited proteolysis with TPCK-trypsin ‘trypsin’ [27,112] or proteinase K [15,26] precisely as described. Following indicated digestion times on the ice, TPCK-trypsin was inhibited by adding a 5-fold excess of soybean trypsin inhibitor (Sigma–Aldrich, Zwijndrecht, The Netherlands). Digestions with proteinase K (Sigma–Aldrich, Zwijndrecht, The Netherlands) were stopped by mixing equal volumes of lysis buffer supplemented with 2 mM PMSF and 2 μg/mL CLAP (chymostatin, leupeptin, antipain and pepstatin (Sigma–Aldrich). Samples were pelleted by 16,000× *g* centrifugation for 5 min at 4 °C, and supernatants were used for immunoprecipitation of CFTR fragments or subjected to thermal denaturation.

### 4.7. Immunoprecipitation

Antibodies against CFTR were incubated with protein-A Sepharose beads (GE Healthcare, Chicago, IL, USA)) for 15 min at 4 °C. Beads were incubated with protease-treated or non-treated lysates at 4 °C for either 3 h or overnight. Immune complexes were washed twice for 15 min at room temperature with 10 mM Tris-HCl pH 8.6, 300 mM NaCl, 0.05% SDS, 0.05% Triton X-100 (for E1-22), 50 mM Tris-HCl pH 8.0, 150 mM NaCl, 1 mM EDTA (for TMD2C), or 10 mM Tris-HCl pH 8.6, 300 mM NaCl, 0.1% SDS, 0.05% Triton X-100 (for G449). Finally, beads were resuspended in 10 μL 10 mM Tris-HCl pH 6.8 containing 1 mM EDTA, followed by an equal volume 2× reducing Laemmli sample buffer, and were thermally denatured for indicated times and temperatures.

### 4.8. Endoglycosidase H Treatment

For some experiments, N-glycans were removed by endoglycosidase H treatment. Solubilized TMD2 translation reactions or immunoprecipitates of proteolyzed full-length CFTR were dissolved in 10 μL 100 mM NaOAc pH 5.4, 0.2% SDS, and incubated at 37 °C. After 10 min, SDS was quenched by adding an equal volume of 100 mM NaOAc pH 5.4, 1% Triton X-100, 1 mM PMSF, 10 µg/mL each of chymostatin, leupeptin, antipain, pepstatin. Samples then received 0.0025 U endoglycosidase H (endo H) and were incubated for 1.5 h at 37 °C. Reactions were stopped by adding 2× reducing Laemmli sample buffer, and the samples were analyzed by 10% SDS-PAGE for TMD2 and 12.5% for CFTR fragments, respectively.

### 4.9. Thermal Aggregation Assay

Lysates of in vitro translation reactions and resuspended immunoprecipitates in reducing Laemmli sample buffer were subjected to thermal aggregation. Aggregation sensitivity of non-immunoprecipitated CFTR (fragments) was determined by exposure for 5 min to 37 °C, 65 °C, 70 °C, or 95 °C, while immunoprecipitated samples from cell lysates were treated for 5 min at 55 °C, or 95 °C. Large SDS-resistant aggregates were removed by centrifugation for 5 min at 20,000× *g,* and samples were loaded onto SDS-PAA gels. The extent of aggregation was determined as the loss of monomeric forms at elevated temperatures and normalized to signals at 37 °C.

### 4.10. Quantitation and Statistical Analysis

The intensity of radioactive bands was measured on a Typhoon ELA-7000 scanner (GE Healthcare Life Science, Chicago, IL, USA)) and quantified using ImageQuantTL software v10.1 (GE Healthcare Life Science). Lane profiles were determined using the same software. Background subtraction on the raw values was performed using the rolling disk with a disk size of 10,000. All graphs were made using GraphPad Prism version 8.0. Student’s *t*-test was used to compare 2 proteolytic fragments, and *p* < 0.05 was considered statistically significant. All conclusions in the manuscript are supported by multiple (>2) independent observations –even when not identical replicates– and strengthened by the internal consistency of all data and figures.

### 4.11. Prediction of Aggregation-Prone Sequences

Aggregation propensity in the unfolded state and folded state were computed as described [44]. The model we used here has served originally as a predictor of amyloid formation but has been applied successfully in alternative contexts [113]. Folding propensities and aggregation propensities often are anticorrelated quantities [44,114]. The amyloid state represents one of the most energetically stable states [115], opposite to the fully unfolded state. The functional physiological state falls between these two extremes, with the fully unfolded state not existing in the cell.

Since the regulatory region R is not resolved in the cryo-EM structure of CFTR [43], minimal and maximal values for accessible surface areas were computed with STRIDE [116] and DSSP library [14] on Alphafold P13569-F1v4.pdb [47], and used to calculate relative solvent accessible surface area per residue. Hydrophobicity plots were generated using the Kyte and Doolittle algorithm [46]. Images of protein structures were based on 5UAK.pdb [43] and created using UCSF Chimera [117] or PyMol 2.5.

## Figures and Tables

**Figure 1 ijms-24-15741-f001:**
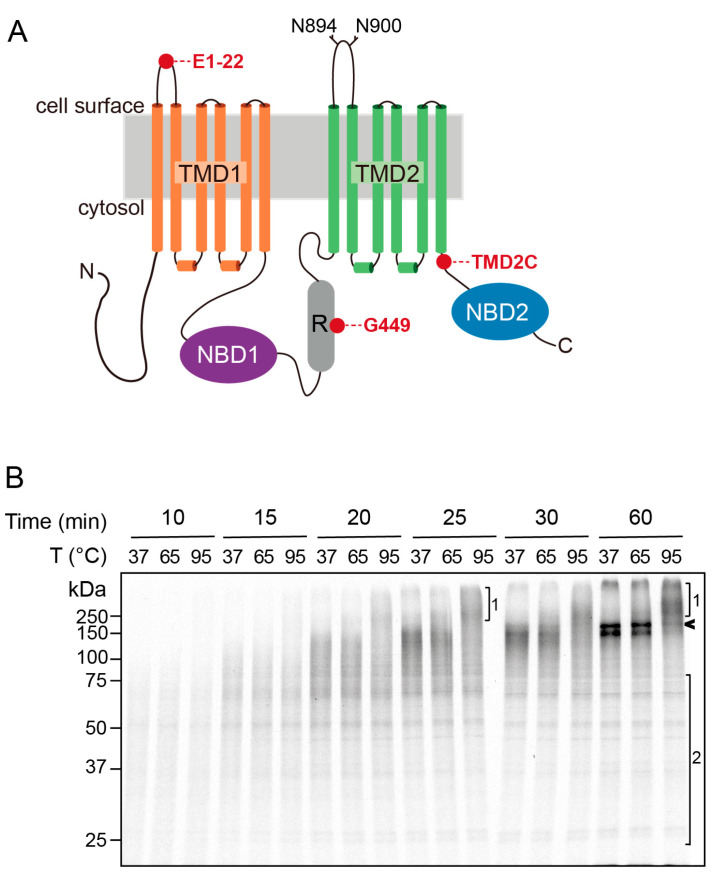
Late CFTR nascent chains are aggregation-sensitive. (**A**). Schematic representation of CFTR’s domain structure. (**B**). CFTR translated and translocated for the indicated times into ER membranes from HT1080 cells was solubilized in lysis buffer and mixed with concentrated Laemmli sample buffer. To investigate the aggregation tendency of CFTR nascent chains, samples were incubated for 5 min at 37 °C, 65 °C, or 95 °C prior to loading onto 10% SDS-PA gel. The arrowhead indicates monomeric full-length CFTR. Bracket 1 shows CFTR aggregates, and bracket 2 indicates CFTR nascent chains that did not aggregate upon 95 °C heat incubation.

**Figure 2 ijms-24-15741-f002:**
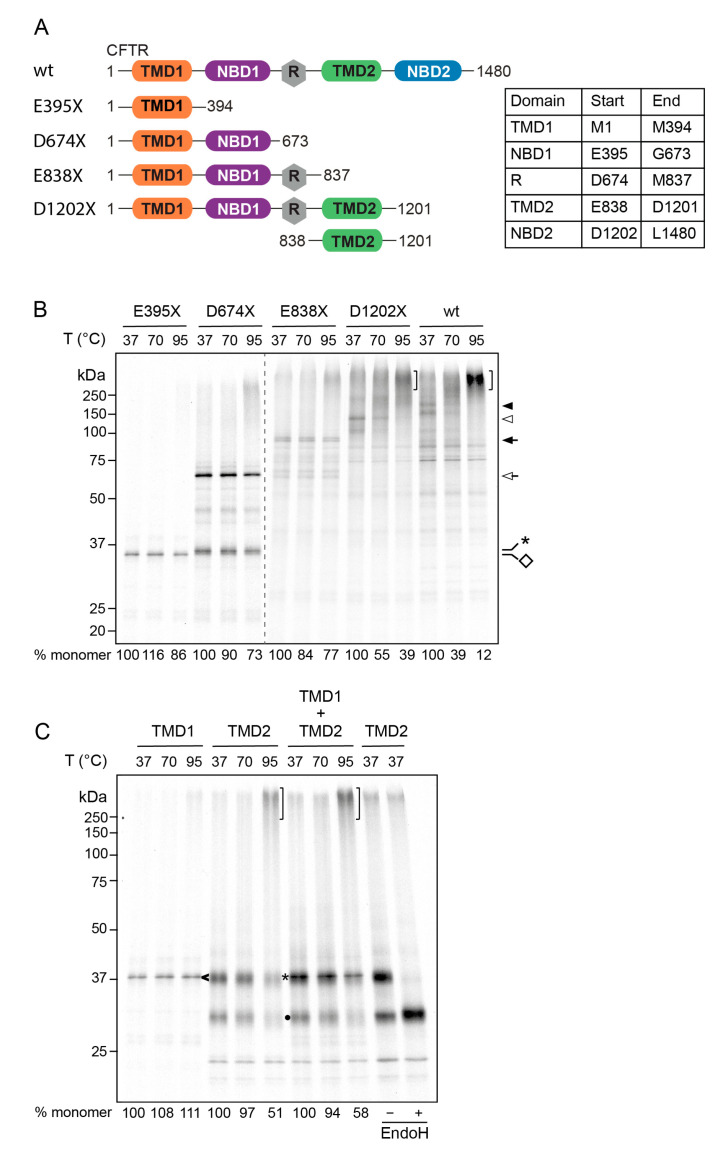
TMD2 confers aggregation sensitivity (**A**). CFTR truncation constructs with the amino acid boundaries in the cartoon and table. (**B**). The aggregation assay was performed at the indicated temperatures as described in Figure 1 on full-length wild-type CFTR and CFTR proteins truncated after TMD1 (E395X), NBD1 (D674X), R-region (E838X), and TMD2 (D1202X). Closed and open arrowheads denote full-length wt CFTR and D1202X, respectively. Closed and open arrows indicate full-length E838X and D674X, respectively, and the diamond denotes full length E395X. Brackets represent protein aggregates and the asterisk denotes D674X nascent chains still attached to the ribosome or prematurely terminated. Percentage monomer is the signal in the monomeric band at a given temperature divided by the signal at 37 °C and is the average of two experiments. (**C**). Isolated TMD1 (E395X) or TMD2 (E838-D1201) were translated in the semi-permeabilized HT1080 cell system, subjected to the aggregation assay as described in Figure 1 and analyzed using 12% SDS-PAGE. The asterisk indicates full-length core-glycosylated TMD2, the filled circle indicates non-glycosylated TMD2, the arrowhead indicates monomeric TMD1, and the bracket indicates aggregates. TMD1 and TMD2 were also synthesized together to investigate their mutual influence. To demonstrate that isolated TMD2 was inserted into the ER membrane, TMD2 samples were digested with endo H to remove N-glycans that are attached during proper translocation into the ER (2 right lanes). Data are representative of two identical biological replicates. The percentage monomer is calculated as in B and represents the average of two experiments.

**Figure 3 ijms-24-15741-f003:**
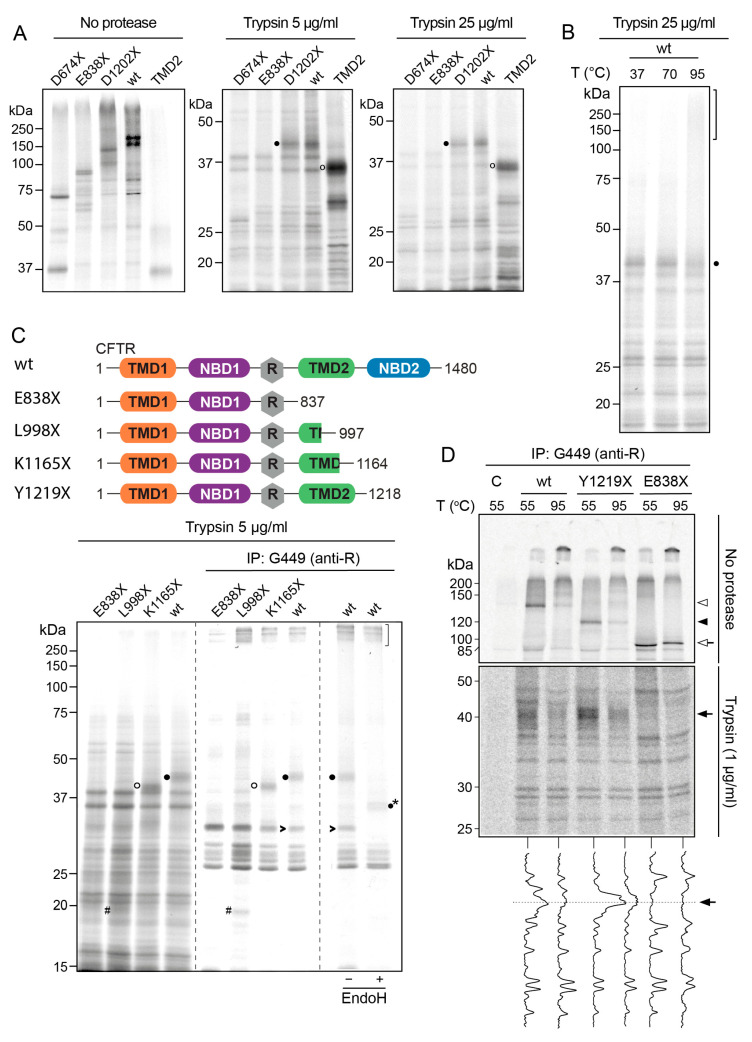
Part of R folds together with TMD2 to form an aggregation-prone structure. (**A**). Indicated constructs (see Figure 2A) were translated in the presence of ER membranes from HT1080 cells. Translocated CFTR variants were solubilized in lysis buffer and digested or not with 5 μg/mL or 25 μg/mL trypsin to determine protease susceptibility. The resulting proteolytic patterns were analyzed by 12% SDS-PAGE. Filled circles indicate a TMD2 protease-resistant fragment found in D1202X and in full-length CFTR. Open circles represent a TMD2 fragment resulting from proteolysis of the TMD2 domain alone. Data are representative of 3 biological replicates. (**B**). To determine the aggregation sensitivity of the R-TMD2 structure, the proteolytic digest of full-length CFTR (in Figure 3A, right panel) was incubated for 5 min at 37 °C, 70 °C, or 95 °C. Samples were then loaded onto a 12% SDS-PA gel to analyze a “precursor-product” relation between a disappearing protease-resistant fragment (filled circle) and accumulating aggregates (bracket). Data are representative of three biological replicates. (**C**). Top: Overview of constructs used. Bottom: Indicated constructs were in-vitro translated in the presence of ER membranes. Translocated CFTR variants were solubilized in lysis buffer and digested with 5 μg/mL trypsin; the proteolytic digests were separated on 12% SDS-PA gel. From these total digests, R-related fragments were immunoprecipitated with the G449 antibody and analyzed on the same gel. R-related fragments indicated by circles include (part of) TMD2. Filled circles indicate the R-TMD2 protease-resistant 42-kDa fragment found in D1202X and full-length-CFTR digests. Open circles indicate this fragment truncated at the C-terminus in TMD2 (derived from K1165X). Hashtag indicates an R-related fragment derived from L998X. Immunoprecipitates of full-length CFTR were treated with endo H to remove N-glycans (most right lane). Glycosylated fragments (indicated by filled circle and arrowhead) shifted down (filled circle with asterisk) or disappeared (arrowhead) upon endo-H treatment. Bracket indicates protease-resistant immunoprecipitated R-TMD2-related aggregates. Data are representative of 2 biological replicates. (**D**). HEK293T cells transfected with indicated constructs were pulse-labeled for 15 min and lysed. CFTR variants were immunoprecipitated from the cell lysates with G449. Immunoprecipitates were heat denatured at 55 or 95 °C and resolved by 7.5% SDS-PAGE (upper gel panel). In parallel, a portion of each lysate was subjected to limited proteolysis with 1 µg/mL trypsin, and protease-resistant fragments were immunoprecipitated with G449. Immunoprecipitates were denatured and analyzed by 12% SDS-PAGE (bottom gel panel). Below this gel are the lane intensity profiles (ImageQuant analysis). Arrows and arrowheads in the upper gel panel indicate the position of the expressed variants. The arrow in the bottom gel panel and lane profiles shows the position of the aggregation-sensitive G449-recognized 42-kDa fragment. Data are representative of two biological replicates.

**Figure 4 ijms-24-15741-f004:**
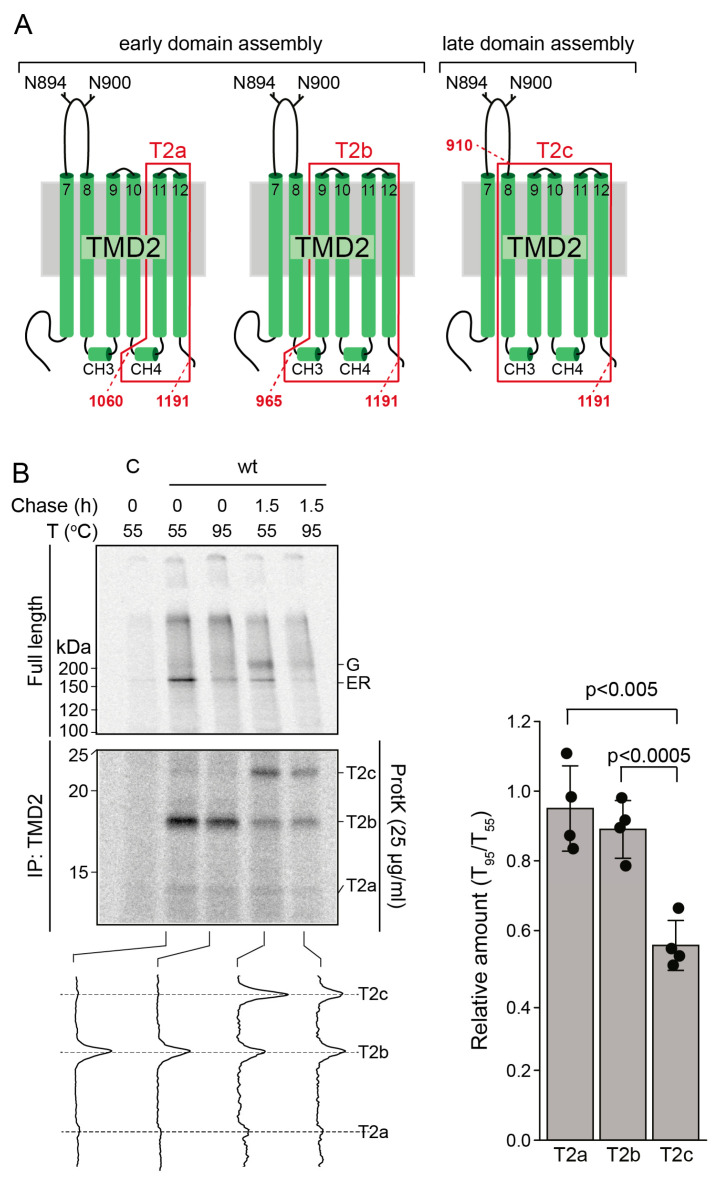
TM7 and TM8 constitute the main unstable determinant in CFTR. (**A**). Schematic representation highlighting the boundaries of the T2a,b, and c fragments generated by limited proteolysis with proteinase K from TMD2 in CFTR [26]. (**B**). HEK293T cells expressing full-length CFTR were pulse-labeled for 15 min and either lysed immediately or after a chase of 1.5 h. CFTR was immunoprecipitated with antibody E1-22 against TMD1 (top panel). The remaining lysates were subjected to limited proteolysis with 25 µg/mL proteinase K, and protease-resistant fragments were immunoprecipitated with TMD2C antibody (middle panel). Immunoprecipitates were heat denatured at 55 or 95 °C and resolved by 7.5% SDS-PAGE (top panel) or 12% SDS-PAGE (middle panel). The bottom panel represents the lane intensity profiles (ImageQuant analysis) of the middle panel. Ratios of T2a, T2b, and T2C at 95 °C versus 55 °C represent means ± SD of four experiments. Student’s *t*-test yielded the *p* values.

**Figure 5 ijms-24-15741-f005:**
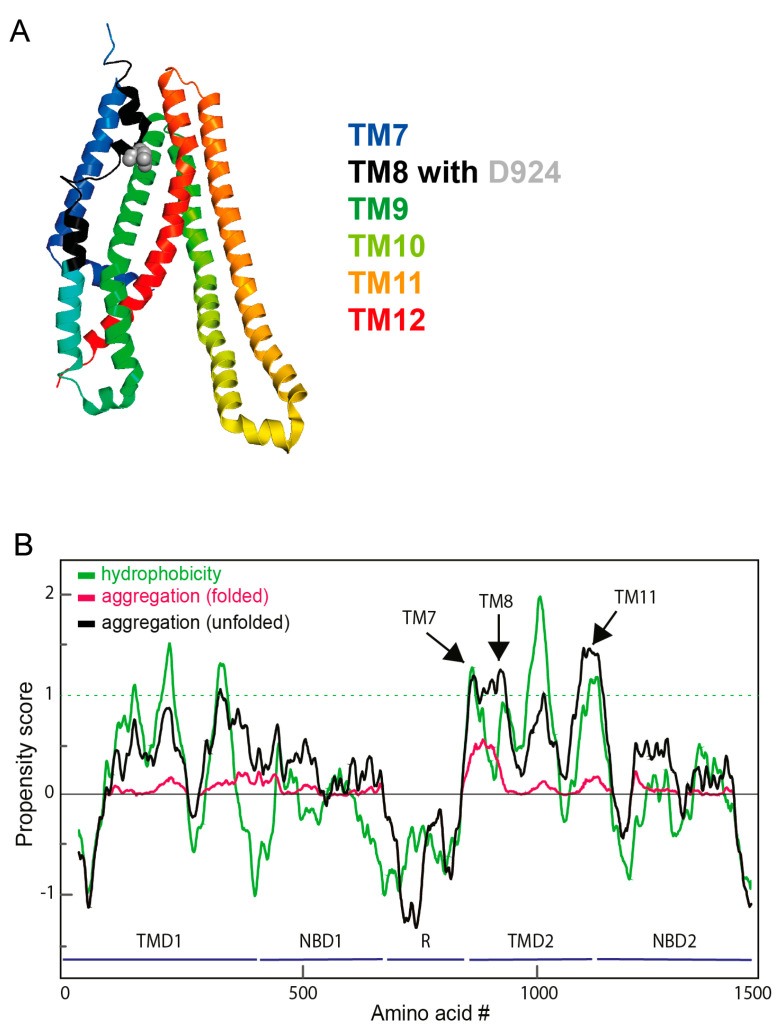
In silico analysis of CFTR aggregation. (**A**). Structure of TMD2, colored from N-terminus (blue) to C-terminus (red) in a rainbow gradient. TM8 (in black) is the only transmembrane helix in CFTR that is discontinuous, with two breaks [43] (PDB: 5UAK). (**B**). Computational bioinformatic analysis of hydrophobicity and aggregation propensity of CFTR. Aggregation and hydrophobicity profile of CFTR. The aggregation profile has been computed using the Zyggregator [44,45] and hydrophobicity [46] algorithms, respectively. Of note, the major peaks of the aggregation-prone region are exactly over the TM7 and TM8 regions.

**Figure 6 ijms-24-15741-f006:**
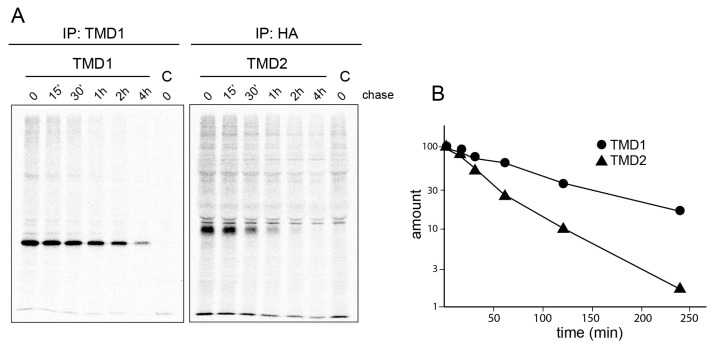
TMD2 is degraded faster than TMD1 in vivo. (**A**). HEK293T expressing individual TMD1 (filled circle) or TMD2 (filled triangle) were radiolabeled for 5 min and chased for indicated periods in the presence of 1 mM cycloheximide to block further translation. Cells were lysed in Triton X-100, and TMD1 and TMD2 were immunoprecipitated with E1-22 and anti-HA antibodies, respectively, and analyzed by 12.5% SDS-PAGE. (**B**). Band intensities at the different chase times were quantitated (ImageQuant analysis) and normalized to the zero chase value. Data are representative of three experiments.

## Data Availability

Not applicable.

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
