# Peer review of "Transmembrane Helices 7 and 8 Confer Aggregation Sensitivity to the Cystic Fibrosis Transmembrane Conductance Regulator"

_ijms, 2023, doi:10.3390/ijms242115741_

Round 1

Reviewer 1 Report

In this manuscript, Kleizen et al provide a detailed characterization of the CFTR regions that contribute to its aggregation sensitivity. The study is relevant, well designed and thorough but there are some points that need to be clarified and justified.

Major concerns

1.     Why wasn’t the aggregation sensitivity of CFTR delTM7-TM8 assessed? This would be relevant for the validation of the hypothesis proposed.

2.     A major issue is reproducibility, which covers several aspects:

-       There is no indication on the number of replicates;

-       In the cases where quantifications are shown, nothing is mentioned regarding how it was performed, nor about statistical analysis, error bars, significance of the differences;

-       In many cases, conclusions are drawn just by simple observation (or absence) of bands. In these cases, where no quantifications are made/possible, original blots (that were not available at all, at least for this reviewer) should be provided for the different replicates performed.

3.     The quality of the figures is very poor – hard to see when printed, and not easier when amplified in the electronic version.

4.     The in vitro translation-translocation assay is quite interesting, but quite old. The authors should consider providing a more detailed description of it in the results description (and not only in the methods).

5.     In most of the experiments, there is no mention which cell line was used as the “backbone”.

Minor concerns

6.     The incidence of F508del is ~70% if you consider the total number of CF alleles, but if you consider people with CF that percentage increases to almost 90% - please correct (line 37).

Reviewer 2 Report

The manuscript entitled «Transmembrane helices 7 and 8 confer aggregation sensitivity to CFTR” by Kleizen et al is of very high quality and of great interest. The work is performed by an excellent group leading in the Cystic Fibrosis field that has made many contributions in the understanding of CFTR biogenesis in the past years. The manuscript is well written and figures convincing.

Using different assays, authors clearly demonstrate that CFTR TMD2 is causing protein instability and that the end of the R-domain is strongly associated with TMD2. These results are precious to better understand CFTR key folding steps and domain assembly processes. The identification of regions difficult to fold or assemble could enable to identify means to rescue mutant CFTR. In this regard, the effect of CFTR modulators on the inherent TMD2 instability would be of great interest. While such experiments will probably be included in a distinct manuscript, this aspect could be discussed in the discussion with a Figure highlighting both modulator binding sites and TM7 and TM8. Also, as this work is in a special issue on "Structure-Based Understanding of the Function-Dysfunction of ABC Transporters", authors could include a short comment to inform readers if such a feature could also occur in other ABC transporters.

The manuscript can be published as is, without any major modifications.

Minor

1. Capital “A” in title: 2.1 aggregation sensitivity of CFTR nascent chain

2. modify ( C ) in the legend of figure 2

3. in figure 5B, the end of the R domain seems to be unstable (black line). If this is the case, this would comfort the assertion that this region contributes to protein instability.

Reviewer 3 Report

Comments to Authors

In this interesting piece of biochemical detective work, Kleizen and colleagues study in depth how variously produced CFTR fragments form protein aggregates when heated in SDS-PAGE sample buffer. The results are clear, experiments are carefully carried out and the evidence largely supports the conclusions (at least with respect to conclusions regarding identification of CFTR fragments responsible for the aggregation phenomenon investigated here).

However, it is not clear to this reviewer how the findings relate to and inform on the physiological folding of CFTR proteins when normally translated in cells. In cells the polypeptides will be continuously sampling folding intermediate conformations and interacting with other parts of CFTR and with chaperones.  In these experiments (as far as I understood) Laemmli buffer will largely denature proteins, disrupting most non-covalent interactions, and the solubilized proteins will be heated to temperatures of 55 or 95 degrees C. The aggregation observed thus relates specifically to these unphysiological conditions.

Below are some examples that caused me some confusion. It is possible that I just misunderstood, and the problems can be rectified by articulating more clearly the experimental procedures and/or the evidence underlying some of the broader conclusions made. Alternatively, the breadth of these conclusions can be qualified.

1. Experiments described in Figure 2C and their description. The situation will be extremely different in in vitro, cell-free translation, and in cells. In cells it is known that proper folding of each TMD involves not only the other TMD, but also the NBDs. For instance the NBD1/TMD “ball and socket joint” (NBD1-ICL1-ICL4) plays a particularly important role in CFTR attaining a native conformation [1, 2] . What physiological relevance can be ascribed to experiments in which aggregation is seen after TMD1 and TMD2 are translated on different ribosomes, the proteins solubilized and then heated? Can anything be concluded regarding the interaction between TMD1 and TMD2 when CFTR is translated in cells?

2. Lines 339 and following “We next assessed aggregation propensity in silico. Using the algorithm in combination with AlphaFold2.0 and the cryo-EM structure of CFTR (PDB 5UAK) confirmed that the major aggregate-prone parts are TM7 and TM8 (Figure 5B), with minor contributions of TM9-11 in the folded structure. In contrast, for TMD1, which has a comparable hydrophobicity profile as TMD2 (Figure 5B), only minor aggregation-prone regions are predicted for the folded state in accord with the experimental data (Figure 2,3).” But wouldn't the "unfolded" prediction be more relevant in analysing results from the experiments performed here (aggregation during heating in Laemmli buffer)?  Yet your experiments do not identify TM11 or elements of TMD1 as important in causing aggregation, despite the presence of major peaks in the aggregation propensity score shown in Figure 5B. Could it be that the algorithm used here is designed to identify sequences likely to cause a different kind of protein "aggregate", a more ordered assembly of molecules (as in amyloid fibrils and beta plaques) than what is likely happening here?

3. Line 232 “showing that the R-TMD2 structure represents a weak point in the stability of CFTR.”; Line 350: “Having established that TMD2 and not TMD1 is the unstable domain in CFTR, we…”; Line 494: “We concluded that TMD2 is the source of CFTR's instability.”; Line 596: “Altogether our results show that C-terminal parts of CFTR in which TMD2 is the major determinant, are more prone to misfolding and aggregation”

I would qualify these statements, which seem too general, as made. In the specific conditions used here, TMD2 appears to be the major determinant for aggregation – but it is likely very different in cells, and it is unclear how the aggregation described here relates to domain stability and misfolding.

Minor points

Line 20 “We conclude that TMD2 extended at its N-terminus with part of the R region forms a stable structure that induces heat instability in CFTR and may be responsible for its limited intracellular stability.” This abstract sentence sounds internally contradictory. Could the wording be changed?

Line 93 “These results showed that C-terminal portions of CFTR are more sensitive to aggregation cq. misfolding than the N-terminal part.” What is meant by “cq.”?

Fig. 2B: Would it be possible to add arrowheads (perhaps on the left edge of the gel, rather than on the gel image?) indicating full-length E395X, D674X and E838X? What is the band at ~35 kDa in D674X?

Line 133 “as described in Figure 2B…” But in the legend for Figure 2B it states "as described in Figure 1".

Line 247: “This implied that the ~39-kDa fragment derived from was a C-terminally truncated version of the 42-kDa fragment derived from D1202X and wild-type. T” Should this be “~39-kDa fragment derived from K1165X”?

Line 254 “(Figure 3C, asterisk” a closing bracket is missing. “An additional immunoprecipitated band of ~30 kDa band was sensitive to endo H as well” One instance of “band” should be removed.

Line 258 “In the digest of L998X truncation that same band must have been a combination of both fragments, one of which then moving up to ~39 kDa and 42 kDa in the digests from the larger CFTR variants (middle panel).” This sentence is unclear.

Line 274: Mention of the TMD1C antibody seems to contradict the figure legend "and protease-resistant fragments were immunoprecipitated with G449", and the figure itself, with a "IP: G449 (antiR)" label.

Line 284 “TMD1 expressed alone…” I don't see TMD1 alone in Figure 3D.

References

1.           Fiedorczuk, K. and J. Chen, Mechanism of CFTR correction by type I folding correctors. Cell, 2022. 185(1): p. 158-168.e11.

2.           Kleizen, B., et al., Co-Translational Folding of the First Transmembrane Domain of ABC-Transporter CFTR is Supported by Assembly with the First Cytosolic Domain. Journal of Molecular Biology, 2021. 433(13): p. 166955.

Round 2

Reviewer 1 Report

The authors answered the aspects raised in quite a peculiar way – several times, evading the questions raised. 

Major concerns

1.     The aspect raised by the authors is relevant, but the same group has described that TM9 functions as a proper insertion signal. In fact, they say “The remaining TM pairs, TM9-10 and TM11-12, each encode signal anchor and stop transfer sequences with short extracellular loops, and it is likely that they insert into the translocon as helical hairpins much like TM3-4 and TM5-6” (Kim & Skach (2012) Front Pharmacol). This should be acknowledged.

2.     The question of reproducibility is relevant in every publication, and it is at least puzzling that the authors try to joke about it and to question reviewer’s intentions. Cloning of a gene is usually followed by sequencing to confirm insertion; generation of an antibody is followed by adequate validation including use of KO for the intended target; determination of a structure does not rely in a single experiment. So, if a result comes from an image from an individual experiment, the least that the authors need to do is to state clearly how many times it was repeated – as they have now done.

To this reviewer, the only information regarding raw data provided in the initial submission was a pdf file containing the sentence “We do not have access now to all the original gel data provided by different co-authors. We will send the original files Monday 4th of September.” Thus, relevant (and requested) information was apparently missing.

Finally, requesting original images from all replicates is already in place in some editorial groups – even if not in MDPI.

All the other questions were properly addressed.
